# The Isolation, Structural Characterization, and Biosynthetic Pathway of Unguisin from the Marine-Derived Fungus *Aspergillus candidus*

**DOI:** 10.3390/md23050219

**Published:** 2025-05-21

**Authors:** Wenjiao Diao, Wei Zhang, Xiaoxi Zhang, Siyu Du, Caijuan Zheng, Xuenian Huang, Xuefeng Lu

**Affiliations:** 1Shandong Provincial Key Laboratory of Synthetic Biology, Qingdao Institute of Bioenergy and Bioprocess Technology, Chinese Academy of Sciences, Qingdao 266101, China; diaowj@qibebt.ac.cn (W.D.); zhang_wei3@qibebt.ac.cn (W.Z.); zhangxiaoxi@qibebt.ac.cn (X.Z.); dusy@qibebt.ac.cn (S.D.); 2Key Laboratory of Tropical Medicinal Resource Chemistry of Ministry of Education, College of Chemistry and Chemical Engineering, Hainan Normal University, Haikou 571158, China; caijuan2002@163.com; 3Shandong Energy Institute, Qingdao 266101, China; 4Qingdao New Energy Shandong Laboratory, Qingdao 266101, China; 5School of Chemistry and Chemical Engineering, University of Jinan, Jinan 250022, China; 6University of Chinese Academy of Sciences, Beijing 100049, China; 7Key Laboratory of Biofuels, Qingdao Institute of Bioenergy and Bioprocess Technology, Chinese Academy of Sciences, Qingdao 266101, China; 8Marine Biology and Biotechnology Laboratory, Qingdao National Laboratory for Marine Science and Technology, Qingdao 266237, China

**Keywords:** heptapeptides, unguisins, *Aspergillus candidus*, biosynthetic pathway

## Abstract

Unguisins, a class of structurally complex cyclic peptides featuring a *γ*-aminobutyric acid residue embedded in the skeleton, exhibit diverse biological activities. Here, a new unguisin K, along with three known congeners, was isolated from the marine-derived fungus *Aspergillus candidus* MEFC1001. The biosynthetic pathway was elucidated through gene disruption coupled with in vitro enzymatic characterization. The *ugs* biosynthetic gene cluster (BGC) containing *ugsA* and *ugsB*, in conjunction with an extra-clustered gene *ugsC*, collaborates to synthesize these unguisins. The alanine racemase (AR) UgsC catalyzes the isomerization of Ala and provides d-Ala as the starter unit for the non-ribosomal peptide synthetase (NRPS). The unique localization of *ugsC* outside the *ugs* BGC is different from previously reported unguisin-producing systems where AR genes reside within BGCs. The methyltransferase UgsB mediates a key pre-modification step by methylating phenylpyruvic acid to yield *β*-methylphenylpyruvate, which is subsequently incorporated as *β*-methylphenylalanine during NRPS assembly. This represents the first experimental evidence of the *β*-carbon methylation of Phe residue occurring at the precursor level rather than through post-assembly modification. The NRPS UgsA recruits a variety of amino acids for assembly and cyclization to form mature unguisins. Additionally, genome mining utilizing UgsA as a query identified homologous NRPSs in diverse fungal species, highlighting the potential for unguisin production in fungi. This study enriches the biosynthetic diversity of cyclic peptides and provides guidance for exploring unguisin-like natural products derived from fungi.

## 1. Introduction

As important producers of bioactive metabolites, filamentous fungi are able to synthesize a variety of compounds with unique structures and diverse functions [1,2,3,4]. Among them, cyclic peptides represent a significant class of metabolites produced by fungi, which possess complex structures and diverse activities [5,6,7,8]. Notable examples include the immunosuppressant cyclosporin A and the antifungal echinocandins [9,10]. Unguisins are a class of cyclic heptapeptides characterized by a high ratio of d-amino acid residues and a *γ*-aminobutyric acid (GABA) moiety. The residue sequence d-Trp-GABA-d-Ala within the scaffold is highly conserved, while other amino acid residues exhibit variability that enhances structural diversity (Figure 1A). To date, ten distinct unguisins have been isolated and identified from several fungal species [11,12,13,14,15,16]. The presence of the GABA fragment confers conformational flexibility to the macrocycle, facilitating interaction with protein targets [17]. Consequently, unguisins exhibit a range of biological activities. For example, unguisin A has been reported as a high-affinity anion receptor for phosphate and pyrophosphate, demonstrating potential applications in environmental and biomedical fields [18].

In recent years, an increasing number of biosynthetic pathways for cyclic peptides have been elucidated in fungi. The basic scaffolds of cyclic peptides are assembled by multi-modular non-ribosomal peptide synthetase (NRPS) utilizing various amino acid residues, as exemplified by the biosynthesis of echinocandin B and FR901379 [19,20]. A recent study on the biosynthesis of unguisins revealed that the *ung* cluster is responsible for the production of unguisins A and B in *Aspergillus violaceofuscus* CBS 115571 (Figure 2A) [21]. Initially, the isomerase UngC, encoded within the *ung* cluster, isomerizes l-Ala to d-Ala, which serves as the starter unit for the heptamodular NRPS UngA. Subsequently, a linear heptapeptide is synthesized by assembling amino acids, followed by the macrocyclization and release from UngA to generate unguisins. Furthermore, a homologous gene cluster, *ung’*, responsible for the biosynthesis of unguisins H and I, was identified in *Aspergillus campestris* IBT 28561 (Figure 2A) [21]. This cluster contains an additional methyltransferase gene, *ungE’*, compared to *ung*. UngE’ is hypothesized to be associated with the methylation of the *β*-carbon of the Phe residue in the cyclic peptide. However, it has yet to be verified whether the methylation occurs as a pre-modification of free Phe or as a post-modification of the cyclic peptide backbone.

In this study, one new unguisin K (**1**) and three known unguisins A (**2**), E (**3**), and F (**4**) were isolated from the marine-derived fungus *Aspergillus candidus* MEFC1001. Based on gene deletion experiments and in vitro enzymatic assays, the biosynthetic pathway of unguisins was elucidated. The alanine isomerase gene is not co-localized with the core heptamodular NRPS gene within the same gene cluster. Furthermore, the methyltransferase mediates the methylation of phenylpyruvate (Ppy), a precursor of Phe, rather than the post-modification of the cyclic peptide. Our study enhances the understanding of unguisin biosynthesis and provides a foundation for further engineered modifications of cyclic peptides.

## 2. Results and Discussion

### 2.1. Isolation and Structural Elucidation of Unguisins from Aspergillus candidus MEFC1001

In our previous study, a novel flavonoid biosynthetic pathway was discovered in *A. candidus* MEFC1001 through genome mining [22]. Continuing our investigation into the secondary metabolites of this fungus, we observed the production of a series of compounds with similar UV absorption in LPM medium. Following large-scale fermentation, compounds **1**–**4** were isolated using chromatographic separation. Comprehensive spectroscopic comparisons revealed that all these compounds are cyclic heptapeptide derivatives containing the non-proteinogenic amino acid GABA (Figure 1B). Based on the HRESIMS and 1D NMR data and by comparison with the reported literature [11,13,14]. compounds **2**–**4** were identified as unguisins A, E, and F, which have been previously isolated from the sponge-derived fungus *Aspergillus candidus* NF2412 [12]. The congener **1** was a new unguisin that was isolated as a white amorphous powder. Its molecular formula was determined to be C_41_H_56_N_8_O_7_ based on the HRESIMS spectrum, which features one additional CH_2_ compared to unguisin A (**2**). The 1D NMR spectral data of **1** are very similar to those recorded for **2**, with the exception of an additional methylene signal (*δ*_H_ 1.23, m, *δ*_C_ 39.2) in **1** and distinct methyl signals (*δ*_H_ 0.60–0.71, d) in **1** versus (*δ*_H_ 0.28, 0.75, d) in **2** in the high-field region. A detailed analysis of the ^1^H-^1^H COSY spectrum revealed that the methylene signal corresponds to the *β*-methylene of a Leu residue (Figure 1C). The HMBC correlations observed from the amide proton of Leu (*δ*_H_ 7.98, d) to the *α*-carbon of Ala-1 (*δ*_C_ 48.1), and from the amide proton of Phe (δ_H_ 8.59, d) to the *α*-carbon of Leu (*δ*_C_ 55.3) provided evidence for a Phe → Leu → Ala-1 sequence (Figure 1C). Consequently, Val-1 in compound **2** is replaced by Leu in **1**, while the sequence of the six identical amino acids in **1** and **2** remains unchanged. The sequence of amino acid residues was further confirmed by the LC-MS/MS fragmentation analysis of **1** (Appendix A). To determine the absolute configurations of amino acid residues, **1** was mildly hydrolyzed and derivatized with Marfey’s reagent. The derivatives were compared with those of authentic amino acids by LC/MS analyses (Appendix A). Accordingly, **1** was elucidated as cyclo-(d-Ala-d-Leu-l-Phe-d-Val-d-Ala-d-Trp-GABA) and designated unguisin K.

The cytotoxic activity of unguisins **1**–**4** was evaluated against the normal human embryonic kidney and liver cells, as well as six cancer cells, using the CCK-8 assays. Compared to the positive control group, none of the unguisins exhibited significant cytotoxic effects on the cell lines at the tested concentration of 50 µM (Appendix A). As a macrocyclic peptide, the activities and applications still require further exploration.

### 2.2. Identification of the Biosynthetic Gene Cluster of Unguisins

To investigate the biosynthetic pathway of unguisins, the annotated genome information of *A. candidus* MEFC1001 was submitted to antiSMASH to predict the biosynthetic gene clusters (BGCs). A total of 41 BGCs were predicted, of which 11 harbored NRPS genes, and only the core gene in the *ugs* cluster encoded a heptamodular NRPS. The core NRPS, designated as UgsA, exhibits 71% and 91% similarity with UngA and UngA’, respectively, which are core enzymes for unguisin biosynthesis in *A. violaceofuscus* CBS115571 and *A. campestris* IBT 28561 (Figure 2A) [21]. To verify its function, *ugsA* was deleted using *hph* as a selectable marker. Compared to the wild type, the mutant Δ*ugsA* lost the ability to produce compounds **1**–**4** (Figure 2B). Therefore, cluster *ugs* was confirmed to be responsible for unguisin biosynthesis in *A. candidus*.

In addition to the *ugsA*, the *ugs* cluster contains 15 other genes encoding typical tailoring enzymes, the functions of which were also characterized individually through gene knockout (Appendix A). The crude extracts of the mutants were analyzed by HPLC (Appendix A). The production of **1**–**4** remained unaffected in all the deletion mutants except for Δ*ugsB*, indicating that only the methyltransferase UgsB is involved in the biosynthesis of these unguisins. In the Δ*ugsB* strain, **3** and **4** disappeared, while compounds **1** and **2** remained (Figure 2B). It demonstrates that UgsB is responsible for the formation of the methyl group at the *β*-carbon of the Phe residue in **3** and **4**.

### 2.3. The β-Carbon Methylation at the Phe Residue Is a Pre-Modification

However, the *β*-carbon methylation of the Phe residue may be a pre-modification of free Phe or a post-tailoring modification of cyclic peptides. To discriminate between these scenarios, recombinant UgsB was heterologously expressed and purified from *Escherichia coli* BL21 and subjected to an in vitro enzymatic assay (Figure 3A). The initial assays utilizing the compounds **1** and **2** as potential substrates failed to generate the methylated derivatives **3** or **4**, conclusively demonstrating that cyclized peptides are not effective substrates for UgsB (Appendix A). It also indicated that UgsB may catalyze the methylation of an extended unit before the NRPS assembly line. The BlastP analysis showed that UgsB exhibits 73.4% amino acid sequence similarity with MppJ, which is involved in mannopeptimycin biosynthesis [23]. MppJ catalyzes the methylation of phenylpyruvate (Ppy) to yield *β*-methylphenylpyruvate (*β*-mPpy), a precursor of *β*-methylphenylalanine (*β*-mPhe). The *β*-mPhe serves as a non-proteinogenic amino acid building block to be incorporated into the NRPS assembly line [23]. Subsequently, Ppy and Phe were tested as substrates, respectively. The LC-MS analysis showed that only Ppy was methylated to produce *β*-mPpy (Figure 3B and Appendix A). Therefore, *β*-carbon methylation at the Phe residue of unguisins is catalyzed by UgsB as a pre-modification prior to cyclic peptide assembly. This contrasts with the hydroxylation of many cyclic peptides, most of which are post-modifications of released cyclic peptides [24,25].

### 2.4. The Extra-Clustered Alanine Racemase UgsC Mediates d-Ala Starter Unit Provision

d-Ala-1, which serves as the starter unit for unguisin cyclic peptide assembly, is derived from the alanine racemase (AR)-catalyzed conversion of l-Ala to d-Ala [21]. As a key enzyme, all AR-encoding genes identified to date are located within the unguisin BGCs, such as UngC in *A. violaceofuscus* CBS 115571 and UngC’ in *A. campestris* IBT 28561 [21]. Notably, when all the genes within the *ugs* cluster were deleted (Appendix A), the enzyme that catalyzes the racemization reaction was not found, suggesting that the AR-encoding gene is located outside the *ugs* cluster in *A. candidus*. A sequence similarity analysis revealed seven racemases that share significant homology with UngC in the genome of *A. candidus* MEFC1001. Among them, the *g_1561* gene encoding protein exhibits 71% similarity to UngC, which is higher than that of the other genes (Appendix A). Meanwhile, the transcript profile of the *g_1561* gene was consistent with those of *ugsA* and *ugsB* under various culture conditions (Figure 4A). The deletion of *g_1561* completely abolished the production of compounds **1**–**4** (Figure 4B), indicating that this gene encodes the functional AR required for unguisin biosynthesis. Therefore, we designate it as *ugsC.*

To further determine its function, recombinant UgsC was obtained using the *E. coli* BL21 expression system (Figure 4C). The enzymatic assay was conducted using l-Ala or d-Ala as the substrate in the presence of pyridoxal 5-phosphate (PLP). After derivatization with Marfey’s FDAA reagent, two Ala derivatives with d and l configurations were detected in the LC-MS profiles (Figure 4D), confirming that UgsC functions as an AR. It provides the d-Ala precursor for the first adenylation domain of UgsA in the biosynthesis of unguisins. Initially, the BlastP analysis ambiguously annotated UgsC as a PLP-dependent transferase rather than an AR. To resolve its evolutionary lineage, we performed phylogenetic reconstruction incorporating three major enzyme classes: PLP-dependent transferases, fungal ARs, and bacterial ARs (Appendix A). Strikingly, UgsC formed a distinct subclade within the PLP-dependent transferase group, showing evolutionary distance to fungal ARs and bacterial ARs. This phylogenetic analysis reveals that UgsC has undergone divergent evolution from both microbial ARs and canonical PLP-dependent enzymes, supporting its classification as a pathway-specific racemase.

### 2.5. Proposed Biosynthetic Pathway of Unguisins and Its Biosynthetic Potential in Other Fungi

Gene deletions combined with the enzymatic assays demonstrated that three enzymes, UgsA, UgsB, and UgsC, are responsible for the biosynthesis of unguisins in *A. candidus* MEFC1001. The biosynthetic pathway of unguisins **1**–**4** was proposed (Figure 5). The extra-clustered AR UgsC catalyzes Ala isomerization and provides d-Ala as the starter unit for NRPS. Additionally, the pre-tailoring enzyme UgsB catalyzes the methylation of Ppy to yield *β*-mPpy, which serves as a precursor for *β*-mPhe that is subsequently incorporated by the M3 module of NRPS as a building block. Within the UgsA assembly line, the bound d-Ala-acyl-adenylate is sequentially extended by l-Leu/l-Val, l-Phe/*β*-mPhe, l-Val, l-Ala, l-Trp, and GABA to form the linear heptapeptide. During this process, the epimerization (E) domains in specific modules catalyze the conversion of l-amino acid residues to their d-configurations. Ultimately, the terminal condensation (C_T_) domain facilitates macrocyclization to yield the final products, unguisins **1**–**4**.

To assess potential fungal producers of unguisin-type compounds, we performed homology-based mining of the core NRPS consisting of seven modules. Nineteen UgsA-homologous NRPSs were identified in public databases based on sequence similarity. Specifically, the phylogenetic analysis focused on the adenylation (A) domain of module 7 (M7-A), which determines the incorporation of GABA residue in the canonical unguisin biosynthesis, while the sequences of A domains that recognize specific amino acids from the previously characterized NRPSs served as the evolutionary controls (Figure 6). The cladogram revealed that the M7-A domains from four reported unguisin-producing NRPSs formed an independent clade. Notably, this clade also includes other species that have not yet been reported to produce unguisins, such as *Penicillium coprophilum*, *Penicilliopsis zonata*, *Aspergillus indologenus*, *Aspergillus unguis*, and *Epichloe festucae*. We propose that these species also have the potential to produce unguisin derivatives. This finding will consequently guide future exploration of fungal-derived unguisin-like natural products.

The yellow background represents the M7-A sequence of nineteen UgsA homologous NRPSs. The red dots represent the strains that have been reported previously to produce unguisins. (SimA accession ID: Q09164; AniA accession ID: AMM63162.1; GliP accession ID: Q4WMJ7.1; Aba1 accession ID: ACJ04424.1; BbBEAS accession ID: ACI30655; CctN accession ID: CRG85572.1; DtxS1 accession ID: XP_007826232.2; and FtmPS accession ID: XP_001261646.1).

## 3. Materials and Methods

### 3.1. General Experimental Procedures

The biological reagents, chemicals, media, and enzymes used in this study were obtained from commercial sources. The primer synthesis and DNA sequencing were carried out by Tsingke Biotechnology Co., Ltd. (Qingdao, China). The codon-optimized sequence of UgsC and UgsB was synthesized by GENEWIZ, Inc. (Suzhou, China). The whole genome sequencing and functional annotation of *A. candidus* MEFC1001 were supported by Novogene Co., Ltd., Beijing, China. The online site antiSMASH (https://antismash.secondarymetabolites.org/ (accessed on 1 November 2021) was used to detect and analyze the gene clusters for potential secondary metabolite biosynthesis in *A. candidus* MEFC1001. The assay of 48 h transcripts in different media was conducted by Novogene Co., Ltd. The unguisin samples were delivered to Qingdao Aike Biotechnology Co., Ltd., Qingdao, China, for the cytotoxic activity test against human non-small cell lung cancer cells (A549), human gastric cancer cells (MKN-45), human cervical cancer cells (Hela), human chronic myelogenous leukemia cells (K-562), human breast cancer cells (MCF7), and human hepatocellular carcinoma cells (HepG2), as well as human normal liver cells (L-02) and human embryonic kidney cells (293T). Cisplatin was employed as a positive control in this cytotoxicity assay.

### 3.2. Strains, Media, and Growth Conditions

The *A. candidus* MEFC1001 strain, employed as the host, was derived from the ocean and was a gift from Naiyun Ji (Yantai Institute of Coastal Zone Research, Chinese Academy of Sciences). Generally, the strain and its mutants were incubated on a potato dextrose agar (PDA) plate at 28 °C for three days for fermentation and protoplast transformation. The preferred medium for the production of unguisins was an LPM medium (9 of g/L glucose, 10 g/L of sucrose, 1 g/L of yeast extract, 1 g/L of peptone, 1 g/L of sodium acetate, 0.04 g/L of KH_2_PO_4_, 0.1 g/L of MgSO_4_, 5 g/L of soybean meal, and 1.5 g/L of CaCO_3_, with the pH adjusted to 6.8 with dilute hydrochloric acid). For the fermentation, 10^5^ fresh spores were collected and inoculated into 50 mL of LPM medium and then incubated at 28 °C and 220 rpm for 7 days. If required, the antibiotic hygromycin was added at a concentration of 50 μg/mL and Geneticin at a concentration of 100 μg/mL. For the RNA extraction, the strain was cultivated at 28 °C, 220 rpm in SGCY medium (2% sucrose, 1% glucose, 0.5% casein hydrolysate, 0.5% yeast extract, 2.1% (*w*/*v*) MOPS, 0.025% K_2_SO_4_, and 1% MgCl_2_·6H_2_O) for 5 days. *E. coli* BL21 (DE3) was cultured in LB medium at 37 °C for the purpose of protein expression. The strains used in this study are listed in Appendix A.

### 3.3. Isolation and Analysis of Unguisins

To analyze the metabolites, the fermentation broth of *A. candidus* MEFC1001 was extracted with an equivalent volume of ethyl acetate and shaken at 25 °C for 1 h. Then, the extract was evaporated by a rotary evaporator and resuspended in 2 mL of methanol to obtain a crude extract for analysis. UPLC (Ultra Performance Liquid Chromatography) with a DAD detector (Waters, Milford, MA, USA) and an Eclipse Plus C18 RRHD column (Agilent, Santa Clara, CA, USA, 50 mm × 2.1 mm, 1.8 μm) was used for rapid analysis. Solvent A was 5% acetonitrile-95% ddH_2_O + 0.05% formic acid and solvent B was 100% acetonitrile + 0.05% formic acid at a flow rate of 0.6 mL/min. The separation program was a linear gradient of solvent B from 0 to 20% over 0.58 min, followed by linearly increasing to 60% at 4.05 min and being maintained for 1.74 min. Finally, it increased linearly to 100% at 6.37 min, then decreased to 0 at 6.95 min and remained at equilibrium for 5 min. The LC-MS analysis was performed using the Agilent 1290 Infinity II and Agilent 6545 LC/Q-TOF systems (Agilent, Santa Clara, CA, USA) coupled with positive and negative electrospray ionization (ESI). The separation procedures and solvents for UPLC were also applicable.

For the preparation and isolation of the products, large-scale fermentation and extraction were carried out. The fermentation broth was extracted three times with equal amounts of ethyl acetate and concentrated to give the crude extract. The product separation was conducted initially using column chromatography under reduced pressure, packed with octadecylsilyl silica gel resin (40 μm, 120 Å). The samples were mixed dry with silica gel and eluted sequentially using 0%, 20%, 40%, 60%, 80%, and 100% methanol solution. The product fractions were pooled in 80% aqueous methanol, recovered, and further purified by semipreparative HPLC. Using a HITACHI device equipped with a C18 preparative column (C18, 5 μm, 150 × 10 mm) and a DAD detector, pure products were acquired by elution with 40% ACN for 30 min, resulting in 4.6 mg of **1**, 16.1 mg of **2**, 11.2 mg of **3**, and 12 mg of **4**.

To detect the structure of the samples, they were dissolved in DMSO-*d*_6_ and the NMR spectra were obtained using a Bruker Ascend Avance IIIHD spectrometer (Bruker, Billerica, MA, USA) at 600 MHz (^1^H)/150 MHz (^13^C). The results are presented in Appendix A.

Compound **1**: white powder; [α]^20^_D_ + 42° (c, 0.01, MeOH); melting point 153 °C; UV (MeOH) λ_max_ 197 (3.56), 219 (4.38), and 279 (3.61); IR (KBr) ν_max_ 3270, 2957, 1643, 1526, 1455, 1384, 1026, and 745 cm^−1^; and HRESIMS *m*/*z* 771.4201 [M-H]^−^ (calcd for C_41_H_56_N_8_O_7_ 771.4199), see Appendix A.

### 3.4. The LC-MS/MS Analysis of ***1***

The LC-MS/MS spectrum was obtained on an Agilent 1290 Infinity II high-performance liquid chromatograph/6545 quadrupole time-of-flight mass spectrometer equipped with a C18 column (Agilent, Santa Clara, CA, USA, 1.8 μm, 2.1 × 50 mm). The gradient elution conditions followed Section 3.3. The collision energy was 50 eV.

### 3.5. Advanced Marfey’s Method to Determine the Absolute Configurations of Unguisin K (***1***)

The advanced Marfey’s method was employed to determine the absolute configuration of amino acids. Compound **1** (3 mg) was dissolved in 4 mL of 6 N HCl, and the mixture was heated at 115 °C for 1.5 h under stirring. The HCl in the hydrolysates was completely removed by nitrogen blow-down and freeze-drying with water addition. The residual hydrolysate and amino acid standards were separately derivatized by adding 200 μL of 1 M NaHCO_3_ and 100 μL of 1% FDAA, followed by incubation at 55 °C for 1 h. The reaction was terminated by adding 100 μL of 2 N HCl. After adding 300 μL of 50% methanol, the derivatized products were analyzed by LC-MS equipped with a C18 column (Agilent, 1.8 μm, 2.1 × 50 mm). The eluent and gradient elution procedures were identical to those described in Section 3.3.

### 3.6. Gene Deletion Experiments

To disrupt the target gene, approximately 1.5 kb of 5′ and 3′ flanking DNA of each gene was amplified from wild-type genomic DNA and fused to the Neo resistance gene or Zhongshengmycin (Nat) from plasmid pJJL-13 by fusion PCR. Each knockout cassette was amplified from the fusion PCR product using nested primers and introduced into *A. candidus* MEFC1001 through polyethylene glycol (PEG)-CaCl_2_-mediated protoplast transformation. The transformants were screened on PDAS plates supplemented with 100 μg/mL of G418 or 1.25 μg/mL of Nat. After being verified for accuracy by PCR, single spores were isolated and purified. Three positive transformants for each deletion mutant were selected for fermentation analysis in the LPM medium. The primer sequences used in this study are listed in Appendix A.

### 3.7. Phylogenetic Analysis

The A domain of the M7 module can be obtained through the following online website: https://nrps.igs.umaryland.edu (accessed on 2 April 2025). The amino acid sequences used in the phylogenetic tree were downloaded from the NCBI database and the multiple sequence comparison was performed using ClustalW 2.1. The phylogenetic tree was constructed using the neighbor joining method based on ClustalW multiple alignment using MEGA 7.0.14 software. The bootstrap values calculated for 500 replications are shown.

### 3.8. UgsC and UgsB Purification and Enzymatic Reaction Assays

The CDS sequences of UgsC and UgsB were codon-optimized and cloned into the pET28a vector to form pET28a-6 × His-UgsC and pET28a-6 × His-UgsB. The recombinant plasmids were then transformed into *E. coli* BL21 (DE3) to express the protein. Positive strains were cultured overnight in 10 mL of LB liquid medium containing 100 μg/mL of kanamycin. All the seed broth was transferred to 1 L of the liquid LB medium containing 100 μg/mL of kanamycin and incubated at 37 °C until the OD value reached 0.6; 0.2 mM isopropyl-D-thiogalactopyranoside (IPTG) was added to the culture and incubated at 16 °C at 180 rpm for 24 h to induce protein expression. The cells were collected by centrifugation and resuspended in 50 mL of lysis buffer (50 mM Tris-HCl, 500 mM NaCl, 5 mM imidazole, and 10% glycerol at a pH of 7.4), then lysed by ultrasound on ice for 30 min. After centrifugation and filtration, the supernatant was loaded onto Ni-NTA affinity chromatography by soft incubation for 2 h. Then, it was washed with 30 column volumes of wash buffer (50 mM Tris-HCl at a pH 8.0, 300 mM sodium chloride, 20 mM imidazole, and 5% (*v*/*v*) glycerol). Finally, the target proteins were eluted with the elution buffer (50 mM Tris-HCl, 500 mM NaCl, 250 mM imidazole, and 10% glycerol at a pH of 7.4). The eluted proteins were concentrated using a 30 (UgsC)/10 (UgsB) KDa Amicon^®^ Ultra-15 device, and desalting was completed using a centrifugal rapid desalting column. The purity of the enzyme was analyzed by SDS-PAGE, and the protein concentration was determined with a micro-volume spectrophotometer.

The enzymatic reaction of UgsC was undertaken in a 50 μL reaction mixture containing 50 mM Tris-HCl buffer (at a pH of 7.5), 2.0 mM l- or d-alanine, 100 μM pyridoxal-5′ phosphate (PLP), and 10 μM enzyme. The reaction was carried out at 30 °C for 1 h and then lyophilized in a freeze dryer. The amino acid derivatization was carried out by dissolving the reaction product in 20 μL of 1M NaHCO_3_, adding 20 μL of 1% FDAA in acetone, and incubating at 55 °C for 1 h. The reaction was terminated by adding 5 μL of 2N HCl. After centrifugation for 10 min, 3 μL of the supernatant was taken for the LC-MS analysis. The substrates l- and d-alanine were also derivatized as controls. In terms of the chromatographic conditions, the chromatographic column used was the same as that in the analysis of the unguisins, and a positive ion mode was adopted for the ESI ion source.

The total volume of the UgsB enzymatic reaction was 100 μL, and it contained 20 μM enzyme, 100 μM substrates (ppy, phe, ugsA (**2**), and ugsK (**1**)), and 1 mM methyl donor S-adenosylmethionine (SAM) in 20 mM Tris-HCl buffer (at a pH of 7.5). The reaction was carried out at 30 °C for 2 h. The reaction was terminated by adding an equal volume of methanol. Subsequently, it was centrifuged at 13,000 rpm for 15 min, and the supernatant was taken for the LC-MS analysis. An ESI source in negative ion mode was used for the analysis. For the column chromatography process, an Agilent SB-Aq column (1.8 μm, 2.1 × 100 mm) was employed, and the gradient elution conditions applied were identical to those utilized in the analysis of the unguisins.

## 4. Conclusions

In summary, one new unguisin, K (**1**), and three known unguisins, A (**2**), E (**3**), and F (**4**), were isolated from the marine-derived fungus *A. candidus* MEFC1001. Their biosynthesis pathway was elucidated based on gene deletions and enzymatic assays. The *ugs* BGC containing *ugsA* and *ugsB*, along with the extra-clustered gene *ugsC*, collaboratively synthesize compounds **1**–**4**. The AR UgsC catalyzes the isomerization of Ala to provide d-Ala as the starter unit for NRPS. Its localization outside the unguisin BGC distinguishes it from previously reported unguisin-producing fungal species, where the AR gene is co-located within the unguisin BGC. The core NRPS UgsA is responsible for the peptide assembly, cyclization, and release. Additionally, the methyltransferase UgsB is involved in the formation of the methyl group on the *β*-carbon of the Phe residue in the scaffold. It acts as a pre-tailoring enzyme that catalyzes the conversion of Ppy to *β*-mPpy, rather than participating in a post-modification process after the cyclic peptide is released from the NRPS. Therefore, one core NRPS, along with two pre-tailoring enzymes, can effectively synthesize these structurally complex natural products. Moreover, apart from the reported producing strains, unguisin-like compounds may also be produced in other fungal species according to the homology analysis. Our work broadens the diversity of cyclic peptide biosynthesis and lays the groundwork for the targeted discovery of fungal-derived unguisin-like natural products.

## Figures and Tables

**Figure 1 marinedrugs-23-00219-f001:**
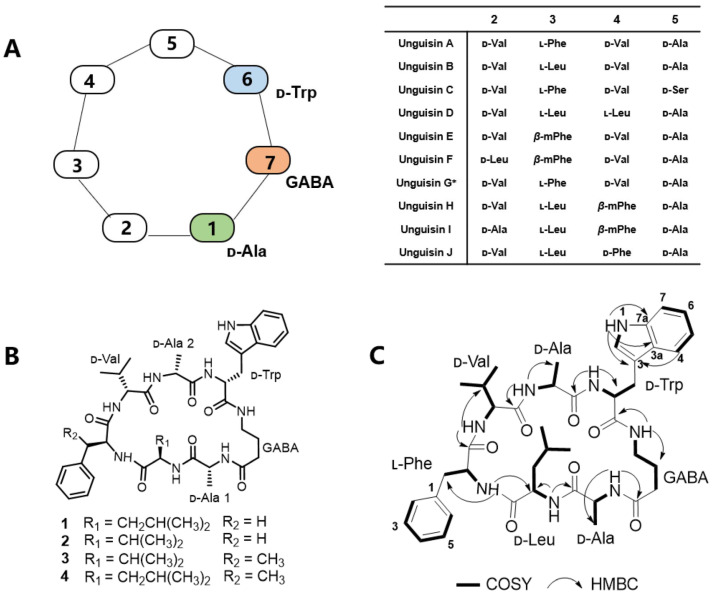
The structures of unguisins isolated from fungi: (**A**) a schematic representation of the unguisins reported previously (* the Trp residue is replaced by kynurenine residue in unguisin G [12]; (**B**) the structures of **1**–**4** isolated from *A. candidus* MEFC1001; and (**C**) the key ^1^H-^1^H COSY and HMBC correlations of **1**.

**Figure 2 marinedrugs-23-00219-f002:**
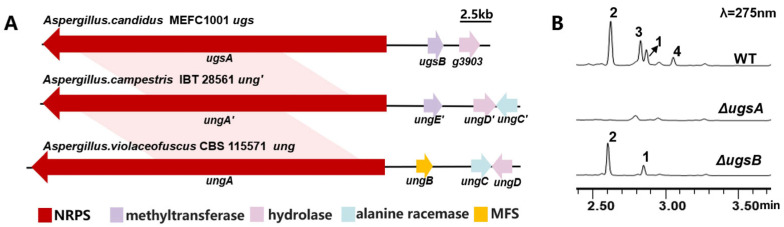
The *ugs* cluster is responsible for the biosynthesis of unguisins **1**–**4**: (**A**) the comparison between two representative unguisins-producing gene clusters and *ugs*; and (**B**) a metabolite analysis of the wild type and gene knockout mutants.

**Figure 3 marinedrugs-23-00219-f003:**
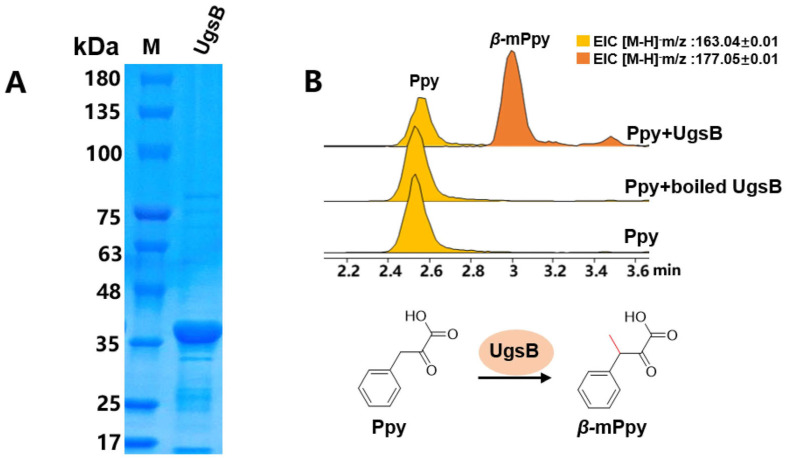
Identification of the function of UgsB through in vitro enzymatic assay: (**A**) purified UgsB protein; and (**B**) in vitro enzymatic assay of UgsB.

**Figure 4 marinedrugs-23-00219-f004:**
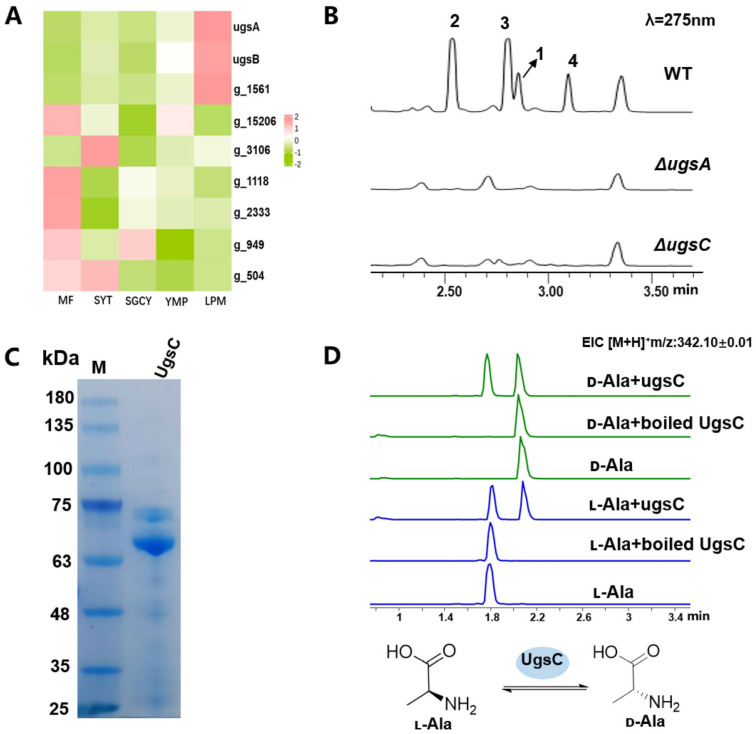
Functional characterization of UgsC: (**A**) transcription levels of alanine racemase-like genes under different culture conditions (MF, SYT, SGCY, YMP, and LPM represent different mediums); (**B**) metabolite analysis of the wild type and mutants; (**C**) purified UgsC protein; and (**D**) in vitro enzymatic assay of UgsC.

**Figure 5 marinedrugs-23-00219-f005:**
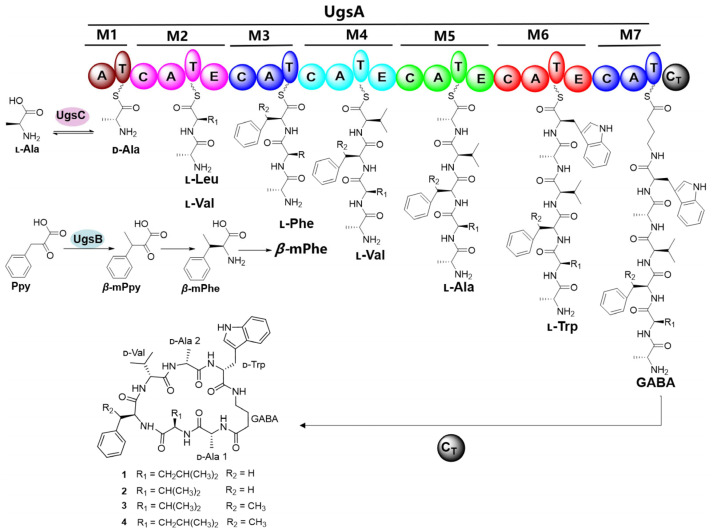
Proposed biosynthetic pathway of unguisins **1**–**4**.

**Figure 6 marinedrugs-23-00219-f006:**
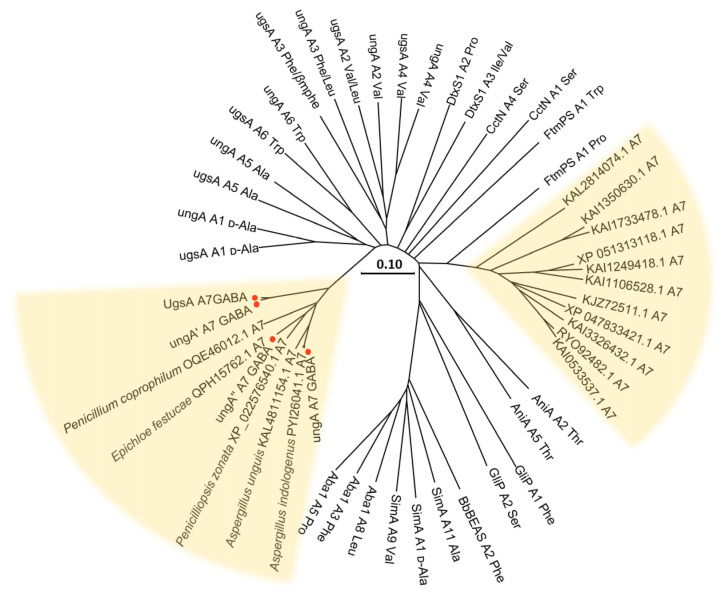
Phylogenetic analysis of the adenylation (A) domain in M7 of NRPS from fungi.

## Data Availability

The data are contained within the article and Appendix A.

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
