# Peer review of "The Isolation, Structural Characterization, and Biosynthetic Pathway of Unguisin from the Marine-Derived Fungus Aspergillus candidus"

_marinedrugs, 2025, doi:10.3390/md23050219_

Round 1

Reviewer 1 Report

Comments and Suggestions for Authors

The authors described discovery of new unguisin derivative and its biosynthetic pathway through recombinant enzyme expression experiments. Finding enzymatic functions in biosynthetic gene cluster is well-designed and it supports systemic biosynthesis of unguisins class peptides. However, the authors determined the absolute configurations of unguisin K via biogenetic relationship. If the absolute configuration of unguisin K is different, the proposed biosynthetic gene cluster should be corrected. Thus, I recommand to perform chemical reaction "advanced Marfey's reaction" for the absolute configurations of amino acids or compare comprehensive genetic analysis between unguisins producer and previously reported gene clusters. Here are my comments on minor errors. 

  1. Please correct all alpha, beta word to italic.
  2. All bacteria's name should be italic. 
  3. 4page, line 154. BmPhe -> B-mPhe

Reviewer 2 Report

Comments and Suggestions for Authors

Manuscript MD3628247, describes the isolation of a new cyclic heptapeptide, unguisin K, along with three earlier described unguisins, from Aspergillus candidus and investigation of its biosynthesis by genome mining, gene knockout mutants, enzyme expression in host organism, isolation of specific enzymes and in-vitro enzymatic assays to prove the identity and products of the enzymes. The structure elucidation of unguisin K seems sound. The gene mining, and study of the enzymatic reactions are significant and prove a unique arrangement of the biosynthetic gene cluster and the unique biosynthetic pathway. I find the manuscript suitable for publication in Marine Drugs although no biological activity was ascribed to unguisin K. However, some typos that continuously appear in the manuscript should be corrected (an annotated PDF of the manuscript is added for you convenience) before the manuscript is accepted for publication:

  • Numbering style of the references in the text and the style of the references in the list of references do not meet the style of the journal.
  • In part of the manuscript the name of bacteria to not appear in Italics.
  • Lines 124-128 – it is better to combine the two sentences by deleting “. They are” and replacing it with a comma “,”.
  • Line 219, Please replace “Meanwhile” with the more appropriate expression: “While”.

Reviewer 3 Report

Comments and Suggestions for Authors

The article concerns the isolation of one new unguisin, complex cyclic peptides featuring a γ-aminobutyric acid residue embedded in the skeleton, unguisin K and three known relative metabolites, unguisin A, E and F from the marine derived fungus Aspergillus candidus MEFC1001. The known metabolites were earlier isolated from other strain of this fungus. The structures were elucidated and identified using comprehensive spectral NMR and mass-spectral data. The authors also have studied the biosynthetic pathways. The pathways were elucidated through gene disruption and in vitro enzymatic characterization. The unguisins biosynthetic gene cluster containing UgsA and UgsB, as well as extra-clustered gene UgsC, works to synthesize these unguisins. The alanine racemase UgsC catalyzes the racemization of Ala in order to provide D-Ala which is the starter unit for the non-ribosomal peptide synthetase. This was elucidated using derivatization of the product of the enzymatic reaction by a chiral reagent followed by HPLC/MS analysis. They also find that methyltransferase UgsB catalises a key premodification step by methylating phenylpyruvic acid to yield β-methylphenylpyruvate. The last is subsequently incorporated instead of β-methylphenylalanine during non-ribosomal peptide synthase assembly. The authors elucidate that β-carbon methylation of Phenyl residue at the precursor is occurred rather than through post-assembly modification. The non-ribosomal peptide synthase UgsA uses a variety of amino acids for assembly and cyclization to form developed unguisins. The authors have carried out a genome mining utilizing UgsA as a query identified homologous non-ribosomal peptide synthases in different fungi and revealed the serious potential for unguisin production in these species.

The article seems to be outstanding by the results, methodology, deeply thinking design of the experiments. I do not any specific notes concerning biosynthetic part of the work.

Nevertheless, the presentation of chemical part of the article seems to be improved. The authors have declared a molecular formula but did not present the certain mass of the deprotonated molecular ion and the calculated mass for such ion. The fragmentation of this quasimolecular ion should be presented as a list of ions with masses and formulae. This should be ultimately presented. The details of the mass-spectrometric experiments should be presented in the experimental part. There are no any physical constants of the new substance – melting point and specific rotation. The specific rotation should be ultimately measured. The data concerning absolute configuration of the aminoacids should be presented. The hydrolysis followed by derivatization of the aminoacids by a chiral agent followed by HPLC/MS is necessary. The declaration concerning biogenetic reasons seems to be not enough because this class of metabolites may contain the aminoacids with different absolute configurations.

There is no information concerning methods of identification of known metabolites. Their spectra should be added to the supporting information.

The title “Identification of unguisin biosynthetic pathway in marine-derived fungus Aspergillus candidus” is not adequate because does not reflect the isolation of new metabolite. It should be changed with something similar to: “Isolation and structure elucidation of unguisin K, a new cyclic peptide and identification of three known relative metabolites from marine-derived fungus Aspergilius candidus MEFC1001 and determination of their biosynthetic ways”.

Note, please, that all the Latin names of living organisms should be presented in italic. Check, please, very carefully entire the text and correct the numerous such mistakes.

The reference 8 should be corrected – Replace, please “Cyclosporine A Biosynthesis” with “cyclosporine A biosynthesis”.

This high scientific level article is exclusively interesting but because of so badly presented chemical information I, unfortunately, have recommend the major revise.

Round 2

Reviewer 1 Report

Comments and Suggestions for Authors

The authors performed proper revision. The article is enough to accept in current form. 

Reviewer 3 Report

Comments and Suggestions for Authors

The authors have adopted all the notes and recommendastions and carried out additional necessary experiments. The manuscript is significantly improved and may be pudlished in current form.